# Perspectives for Ezrin and Radixin in Astrocytes: Kinases, Functions and Pathology

**DOI:** 10.3390/ijms20153776

**Published:** 2019-08-02

**Authors:** Amin Derouiche, Kathrin D. Geiger

**Affiliations:** 1Institute of Anatomy II, Goethe-University Frankfurt, D-60590 Frankfurt am Main, Germany; 2Neuropathology, Institute for Pathology, Carl Gustav Carus University Hospital, TU Dresden, D-01307 Dresden, Germany

**Keywords:** radixin, ERM, phosphatases, mGluR3, mGluR5, PKCε, GRK2, astrocytoma, glioblastoma, epilepsy, glial activation

## Abstract

Astrocytes are increasingly perceived as active partners in physiological brain function and behaviour. The structural correlations of the glia–synaptic interaction are the peripheral astrocyte processes (PAPs), where ezrin and radixin, the two astrocytic members of the ezrin-radixin-moesin (ERM) family of proteins are preferentially localised. While the molecular mechanisms of ERM (in)activation appear universal, at least in mammalian cells, and have been studied in great detail, the actual ezrin and radixin kinases, phosphatases and binding partners appear cell type specific and may be multiplexed within a cell. In astrocytes, ezrin is involved in process motility, which can be stimulated by the neurotransmitter glutamate, through activation of the glial metabotropic glutamate receptors (mGluRs) 3 or 5. However, it has remained open how this mGluR stimulus is transduced to ezrin activation. Knowing upstream signals of ezrin activation, ezrin kinase(s), and membrane-bound binding partners of ezrin in astrocytes might open new approaches to the glial role in brain function. Ezrin has also been implicated in invasive behaviour of astrocytomas, and glial activation. Here, we review data pertaining to potential molecular interaction partners of ezrin in astrocytes, with a focus on PKC and GRK2, and in gliomas and other diseases, to stimulate further research on their potential roles in glia-synaptic physiology and pathology.

## 1. Introduction

### 1.1. Astrocytes

Astrocytes are a major cell population present throughout all regions in the CNS. Over the past twenty-five years, the understanding of the astrocyte has shifted from a passive cell type supplying structural and metabolic support to neurons, such as ion homeostasis, neurotransmitter clearance, and energy supply, to an active player in cell-to-cell communication. Thus, astrocytes participate in the maintenance of the blood–brain barrier and neurovascular coupling [1]. Observations that in addition to neurons also astrocytes can release glutamate [2] have further developed to the concept of multi-faceted glia-synaptic interaction [3]. Research into its functional significance in the brain has recently revealed that astrocytes play important roles in physiological brain circuits, functions in naturally occurring behaviours [4], and rewiring in pathology [5]. The structural correlation of the glia–synaptic interaction are the peripheral astrocyte processes (PAPs [6]). These processes extend from the glial main processes, which contain glial fibrillary acidic protein (GFAP) positive glial filaments, and can be very small, down to the dimensions of pure membrane appositions (50–100 nm). In the grey matter, they permeate the neuropil and contact defined surfaces of neurons [7]. In particular, many PAPs are structurally closely related to synapses, although this relation is not systematic [8]. In their entirety, the PAPs constitute 70–80% of the astrocyte’s cell surface [9]. Importantly, membrane-bound proteins important in cell-to-cell, often glia-neuronal interactions are preferentially localised if not restricted to the PAPs (for example CD44, see below). Based on these structural and biochemical properties the PAP can be regarded as a cellular compartment [6,10]. Much of the glia-synaptic interactions underlying the glial role in physiological functions [4,11,12] are accomplished by the PAP compartment. As an important property of this astrocyte compartment, we have shown that it is selectively visualised by ezrin or radixin, two members of the ezrin radixin, moesin (ERM) family of proteins [13].

### 1.2. Ezrin in Astrocytes 

Probing astrocytes for expression of ERM proteins was initially inspired by the astonishingly small dimensions of the PAPs [13], and the idea that this might be based on molecular mechanisms similar to those of microvilli. Since their discovery, ERM proteins were considered microvillus proteins involved in structural maintenance, or in motility mechanisms of actin-based surface protrusions. A systematic study of ERM protein distribution, thus, includes virtually no neural tissues, apart from ependymal epithelial cells [14]. The general consensus was that ERM proteins are normally enriched in apical subdomains of microvilli-bearing surface epithelia such as the placenta, intestinal mucosa, or renal ducts [15] or highly dynamic cells (lymphocytes [16]). The finding of moesin in endothelial cells in many organs, including the brain [14] is in line with the role of ERM proteins in establishing and maintaining extremely narrow membrane appositions. This functional role is further reflected by the localisation of ezrin and radixin in astrocytes, especially since they are preferentially in the PAPs [13]. 

ERM proteins can establish such membrane appositions, such as for example in microvilli by tethering the cell membrane to the central actin filaments [17,18]. They have a binding site for membrane-bound proteins near the N-terminus-the so-called FERM-domain-which binds to small GTPase regulators, cell adhesion proteins, or adapters, and another one at the C-terminus that can bind to actin or the FERM domain. ERM proteins can assume a ‘closed’ conformation owing to a central hinge region, and be inactive through self-inhibition, by mutual binding of the C- and N-terminal regions. Phosphorylation of the conserved C-terminal threonine (T567 in ezrin, T564 in radixin and T558 in moesin) is required for activation, ‘opening’ of ERM proteins. The membrane-to-cytoskeleton linker function of ERM proteins is related to this site. Notably, there are several other phosphorylation sites, and ERM proteins can engage in protein interactions and functions completely different from that of membrane-to-cytoskeleton linking [17,18]. For example, ERM proteins may also be crucial in stiffening of the actin cell cortex or in the assembly of receptor complexes [18], but to the best of our knowledge, there are no studies as to which role(s) ERM proteins play in the fine astrocyte processes. 

The conformational switch of ezrin from the dormant to the active form is complex [17,19]. The phosphorylation of the C-ter threonine although necessary is not sufficient [20]. The interaction of ezrin with phosphatidylinositol, 4, 5-bisphosphate (PIP_2_) is also important for activation. In a two-step model of activation, ERM proteins are recruited to membrane regions rich in PIP_2_, possibly rendering the threonine amino acid residue more accessible to phosphorylation [20,21].

In the astrocyte, ezrin is required for filopodia formation in cell culture, and involved in the brain, in PAP motility [22]. In general, ezrin maintains the mechanical cohesion of membranes with F-actin, thus facilitating actin-related cellular morphological changes [23]. It can tether membranes, as in microvilli. However, in contrast to microvilli astrocyte processes in the brain are highly irregular structures. In this context, it is helpful to consider that ezrin can be targeted to curved membranes, depending on membrane conformations and interactions with actin, and curvature-sensing binding partners [23]. For example, ezrin is enriched in negatively-curved membrane protrusions through interaction with curvature-sensing inverse-Bin-Amphiphysin-Rvs (I-BAR) domain proteins [23] (see below). 

While the preferential localisation of astrocytic ERM proteins in the PAP attracts much interest in the context of glia-synaptic interaction, ezrin and radixin are also present in the fibrous astrocytes occurring in white matter (rat [13]), which is devoid of synapses. Notably, ezrin and radixin staining intensely label soma and stem processes of fibrous astrocytes, thus visualising the complete astrocyte shape [13]. Obviously, ERM proteins in this localisation do not contribute to fine membrane protrusions, and their functional role remains unclear. Although this has not been investigated, the ERM proteins in soma and stem processes might be present in their non-phosphorylated, inactive form, as is the case in cultured or in grey matter astrocytes, where phospho-ezrin is almost restricted to filopodia or PAPs, respectively [22]. Phosphorylated ERM proteins in fibrous astrocytes might then be expected to be present in their finger-like processes extending to the axon, at the node of Ranvier [24] (see below). ERM protein expression is not down-regulated, and it remains to be investigated whether these abundant inactive molecules represent a reserve pool (also in protoplasmic astrocytes) or fulfil other functions independent of phosphorylation (see below).

Regarding pathology, a striking feature is the ezrin-staining of vessel-associated astrocytes in severe encephalopathy. It appears that vascular astrocytes can be of either subtype, fibrous or protoplasmic [25], but it is currently unclear how A1 and A2 reactive astrocytes relate to subtype. However, when associated with deleterious changes, strong ezrin expression often comprising the whole cell may be correlated with astrocytes directly contacting blood vessels. Similar alterations can be found surrounding malignant glioma.

Process motility in astrocytes can be stimulated by the neurotransmitter glutamate, through activation of the glial metabotropic glutamate receptors (mGluRs) 3 or 5 [22]. It has, however, remained unclear how the mGluR stimulus is transduced to ezrin activation, or what are the kinases for the C-ter threonine phosphorylation. Clearly, identifying the astrocytic ERM-activating enzyme(s), and membrane-bound binding partners of ERM proteins that possibly link to the extracellular matrix or other cells, will open new approaches and understanding of the glial role in physiological and higher brain functions. Apart from process motility and membrane-to-cytoskeleton bridging, ERM proteins function in completely different contexts in non-neural cells, such as cell division, assembly of receptor complexes, cellular polarisation, or stiffening of the cell cortex [18] (see below). Together with the possibility that astrocytic ezrin and radixin might also be non-redundant and play different roles, this leaves many open questions. Thus, the current understanding of ERM proteins in astrocytes is only beginning to evolve. To stimulate further studies we review here the involvement of ezrin and radixin in astrocyte functions. We provide indications for glial ERM bindings partners, kinases collected from work on astrocytes but also on cell lines, with a view to normal brain function and possible roles in pathology.

### 1.3. Other Ezrin Expressing Cell Types in the CNS 

Since by western blot of the brain only weak ezrin expression was detected [26,27] and immunostained CNS parenchyma was negative [14], only ependymal cells were initially considered ezrin positive [14] (for review see Reference [27]). As a present view, astroglial cells (including ependymal and retinal pigment epithelial cells) are the only ezrin-expressing cells in the normal, adult mammalian CNS, with the exception of restricted expression in relation to adult neurogenesis (see below, [28]). Astrocytes also express radixin [13], which is also found in interneurons where it is essential for organising GABA receptors [29].

During development, ezrin appears in CNS axon terminals of select neuronal subtypes and highlights the corresponding sublayers in laminated terminal fields. Within the chick retina, the only ezrin positive structures are the cholinoceptive ganglion cells, and their specific sublamina in the optic tectum, where all retinal ganglion cells project [30]. Similarly, ezrin is restricted to the cutaneous sensory neurons within the chick dorsal ganglion, and its specific dorsal horn sublamina [30]. Whereas diffuse staining of neuronal somata (and neuropil) is also a general observation in rat hippocampus during the first postnatal week (Derouiche, unpublished), this is not reflected in the adult, and there is also no neuronal staining in the adult mouse retina [10,31]. 

The presence of ezrin in neuronal growth cones in cell culture, in particular in their filopodia (Derouiche, unpublished data [30,32]) may be related to labelling of the terminal fields mentioned, where arriving fibres have to search for and establish appropriate synaptic contacts. It remains unclear why the axon terminals of the projections mentioned above maintain ezrin expression after synaptogenesis. One of the roles of ezrin in the neuronal growth cone may be the formation of its filopodia, which are highly motile, actin-based, and devoid of microtubules. Interestingly, ezrin in developing, cultured neurons also enables neuritogenesis through down-regulation of RhoA activity [33].

At the nodes of Ranvier, the glial contacts are supplied by astrocyte finger-like processes (‘coronae’ [24]), they can be assumed to contain the astrocytic ERMs, ezrin and radixin, but this has not been explicitly demonstrated. Although in the peripheral nervous system, it is interesting to mention in this context the microvillus-like Schwann cell processes, which contact the axon at the nodal region [34,35,36]. These processes express all three ERM proteins in abundance ([35,36], for review see Reference [28]), and experimental paranodal disruption involves calpain-induced ezrin loss in microvillus-like Schwann cell processes [37]. 

Ezrin also appears to play a role in adult neurogenesis, the regeneration of neurons in the mammalian CNS. This occurs in the subventricular zone, i.e. in the wall of the forebrain ventricles, and the subgranular zone of the dentate gyrus of the hippocampus. As a morphological feature of the subventricular zone, the actual neuronal stem cells are sequestered from the neuropil by a layer of process-bearing cells with astrocyte-like properties. The subventricular zone also generates the neuronal precursors replacing interneurons in the olfactory bulb. Their migration to the olfactory bulb along the rostral migratory stream is guided by the glial tube, cord-like bundles of astrocytic somata and processes. Glial tube astrocytes display a bipolar morphology, extending processes in opposite directions along the RMS in parallel with each other to form a cord-like bundle. Ezrin is expressed by neural stem cells of the subventricular zone as well as the neuroblasts and astrocytes but not the oligodendrocytes originating from them [38]. In the rostral migratory stream, ERM proteins are differentially expressed, with radixin and moesin in neuroblasts, and ezrin in astrocytes of the glial tubes [39]. In the subgranular zone of the dentate gyrus, radixin was detected also in young neurogenic stem cells [40] and in oligodendrocyte precursor cells, which have radixin also in many other brain regions [40], and are known to migrate over long distances. It will be interesting in this context of adult neurogenesis to study ERM proteins for possible functions in cell division or migration, in view of the known ERM functions in development [18].

## 2. Potential Mechanisms of Ezrin Regulation in Astrocytes 

### 2.1. Upstream Signals of ERM Activation

There is only scarce evidence on upstream signals of ERM activation initiated by stimulation of membrane receptors. In astrocytes both, in culture and in the brain, phosphorylated ezrin preferentially localises to filopodia and peripheral processes, respectively, especially around synapses, and filopodia formation is mediated by mGluR3 or 5 stimulation [22]. Although the direct mGluR3 or 5 dependent ezrin phosphorylation has not been tested [22], this indicates that ezrin phosphorylation may also be mediated by mGluR3 or 5 stimulation. This is supported in astrocytes by the induction of ezrin phosphorylation by the G protein-coupled receptor (GPCR) kinase (GRK) GRK2, which is coupled to mGluR5 [41], but again a direct link between mGluR5 stimulation and ezrin phosphorylation has not been demonstrated [41]. Since both results [22,41] were obtained in astrocytes, they converge to a working hypothesis for extension of PAPs around synapses, as modelled experimentally by glutamate-induced filopodia formation. Thus, stimulation of astrocytic mGluR3 or 5 leads to activation of GRK2, the ERM kinase directly phosphorylating ezrin and/or radixin, which initiates process remodelling. Further studies will have to test whether mGluR3 or 5 stimulation, in fact, leads to enhanced ezrin phosphorylation, and GRK2 is preferentially localised in astrocytic filopodia, or PAPs in the brain. 

Ezrin function is linked to receptor stimulation also in a very different context. It has been shown in a non-neural cell line that β2 adrenergic receptor stimulation leads to receptor internalisation, and this is mediated by GRK activity and blocked by inhibition of ezrin function [42]. Here, ezrin may function in a way independent of its membrane-to-cytoskeleton linker function [18].

In addition, sphingolipid signalling is established to activate ERM proteins in a wide variety of cells [43]. Although astrocytes display pronounced sphingolipid signalling, which may also be of medical importance [44,45], a link to ERM phosphorylation remains to be studied in astrocytes.

### 2.2. Binding Partners

Much research has focussed on further elaborating the interactions of ERM proteins with membrane-bound proteins such as CD44, CD43, ICAM-1 and ICAM-2, mostly adhesion proteins, as first established by Heiska et al. [46] and Yonemura et al. [47]. Of these, CD44 is best studied, since it is the main receptor protein for hyaluronan, the main constituent of the extracellular matrix in the CNS. CD44 is expressed by a subset of astrocytes in grey and white matter [48]. While the subcellular juxtaposition of ERM proteins and CD44 in astrocytes awaits demonstration, ezrin/radixin bridges between the actin cytoskeleton and membrane-bound CD44 might be key steps in the formation of astrocytic filopodia. Konopka and co-workers [49] recently have shown that treating cultured astrocytes with hyaluronidase or knocking down CD44 results in astrocyte stellation, whereas overexpression of CD44 causes cell flattening. These CD44-mediated changes in astrocyte morphology can be blocked by inhibition of Rac1 activity [49]. Ezrin or radixin were not considered in that study [49], however, their participation in the effect cannot be excluded in the light of ICAM-2 signalling. ICAM-2 has not been detected in astrocytes. However, interestingly, the results obtained from endothelial cells (human umbilical vein endothelial cells, HUVEC) suggest that ICAM-2 mediates activation of Rac1 via a signalling pathway involving ERM proteins [50]. ICAM-1 is present in human astrocytes after cytokine induction and under pathological conditions [51,52], but the possible interaction with ezrin has not been studied in these cells. CD43, another membrane-bound ERM interacting protein [47], is a leukocyte antigen not detected in astrocytes.

### 2.3. Kinases

While the mechanisms and conditions of ERM activation appear universal at least in mammalian cells and have been studied in great detail (see above), the actual ezrin and radixin kinases appear cell type specific, and may be multiplexed within a cell. It is also feasible that given diverse functions of ezrin [18] there might be different ERM activating enzymes in different cellular compartments, or at different phases of cellular development or activation. A list of kinases known to activate ezrin in various cells has been compiled by Adada et al. [43], however, there is little information on cells of the nervous system. Below, we will summarise the sparse evidence for enzymes phosphorylating ERM proteins in astrocytes, and review data from other cells that might appear promising to study in astrocytes.

#### 2.3.1. PKCs

Since PKCs generally phosphorylate serine or threonine residues [53], they have been considered as potential activators of ERM proteins at the crucial C-ter threonine phosphorylation site. In addition, PKCs can respond to second messengers and phosphorylate a wide variety of downstream targets in a tissue and cell type specific way, placing them at the crossroads of membrane receptor signalling and integrated cellular responses to physiological or pathological stimuli. The PKC family of proteins consists of three groups [53]. The conventional PKCs (α, βI, βII, γ) are Ca^2+^ dependent, and PKCγ also senses oxidative stress (oxidative or reactive oxygen species) by oxidation of cysteine residues. The novel PKCs (δ, ε, η, θ) do not require Ca^2+^. Conventional and novel PKCs have a DAG sensor. The atypical PKCs (ζ, ι) have different forms of activation and are Ca^2+^ independent. Analysing the PKC expression profile in primary astrocytes, Slepko et al. [54] found an absence of PKCγ and the presence of all other isoforms with a high expression of PKCε. Several isoforms have been assayed for their capability of ERM protein phosphorylation in cultured cell lines, however, only one, PKCε, directly in primary astrocytes (see below). 

Regarding the downstream effects of PKC isoforms on ERM proteins, recombinant human PKCα can phosphorylate ezrin in a cell-free assay in a Ca^2+^ and phospholipid dependent manner, and PKCα can phosphorylate the ERM C-terminal threonine residue within a kinase–ezrin molecular complex, in breast cancer epithelial cells [55]. PKCθ phosphorylates the C-ter threonine site in moesin, in a cell-free assay and in leukocyte extracts [56]. In intestinal epithelial cells, which are highly polarised displaying apical microvilli, ezrin is activated by the atypical protein kinase Cι [19]. PKCδ is an isoform highly expressed in astrocytes but has not been tested as an ezrin kinase. The activity of PKCε, which is also highly expressed in astrocytes and is the only isoform directly tested for ERM protein activation in astrocytes, was found to lead to filopodia formation [57]. Interestingly, PKCε in primary astrocytes phosphorylates radixin but not ezrin [57]. However, astrocytic ERM proteins were not detected as PKC interaction partners in a proteomic analysis [58]. Altogether, of the PKC isoforms present in astrocytes, PKCα, PKCθ, and PKCι have been shown to activate ERM proteins in cell-free assay or other cell types, and PKCε does so in astrocytes.

#### 2.3.2. Small GTPases 

It is firmly established by a large body of literature that the rho family proteins, particularly rhoA, cdc42, and rac1 are key factors in dynamic re-organisation of the actin cytoskeleton (for recent reviews see References [59,60]). 

In addition, cultured astrocytes are well-known to express the rho family proteins rhoA, cdc42, and rac1, and small GTPases of the rho family have been shown to control changes in astrocyte morphology both, in cell culture and in the brain (reviewed by Reference [61]). In the normal, unlesioned brain a role of astrocytic ERM proteins ezrin and radixin may be to adapt the plasma membrane to the cytoskeleton [18], and thus to establish and maintain the narrow structure of PAPs [22]. PAP motility, as occurs in activity-associated glial plasticity [22], also entails re-arrangement of the actin cytoskeleton, and elongation of the actin core bundle in the PAPs, a processes not accomplished by the ERM proteins. ERM proteins and rho family proteins may be conceived to cooperate in process motility in the perisynaptic environment. Since ezrin and radixin play a role in cell shape changes similar to those of the rho family GTPases, ERM proteins have been considered as downstream targets of rho family proteins [62,63]. However, since the first demonstrations of rho family proteins in cultured astrocytes [64,65], these proteins have been associated with glial activation and cell migration, with a view to glial scar formation and pathological stimulation. In line with this, rho family protein expression is low in astrocytes in the brain [61], much lower than in neurons [66] but considerable in astrocyte culture. In tissue, the astrocytic expression of rho family GTPases is linked to glial activation such as after spinal cord lesion [67].

In glial activation in cell culture, it has been shown that ERM proteins and rho GTPase can act synergistically to execute cell shape changes [68], but rho GTPase does not activate ERM proteins. Deactivation of the rho GTPase-ROCK (Rho kinase) pathway and phosphorylation of ERM proteins are both triggered by IL-1β [68]. However, ERM phosphorylation is mediated by a rho-ROCK-independent pathway [68]. This indicates that both pathways are separate but can be orchestrated by upstream signalling (IL-1β) and that there is no molecular interaction of rho GTPase/ROCK and ERM proteins. This is in line with other reports investigating mechanistic links between ERM proteins and the small cytoskeleton-associated GTPases. Thus, although rho-kinase and ROCK were found to phosphorylate the C-terminal threonine residue of the ERM proteins in a cell-free assay [55,62,63], the same group later showed that phosphorylation at this ERM site is, in fact, insensitive to ROCK inhibition in cultured cells (NIH3T3 or HeLa cells [69]). In line with this, the inhibition of rho kinase in cultured cells does not prevent ezrin phosphorylation and membrane ruffling (Hep2 cells, a human cancer cell line, [42]. In cultured astrocytes, ERM proteins on an appropriate stimulus can form a complex with NADRIN and Na^+^/H^+^ exchanger regulatory factor (NHERF1). This complex, in turn, inactivates RhoA and leads to astrocyte stellation, i.e., change from flat to process-bearing morphology [70]. Notably, this rare case of an ERM-GTPase interaction is a signalling pathway not including the actin-to-membrane binding function of ERM proteins. Altogether, in spite of reports supporting mechanistic links between the small cytoskeleton-associated GTPases and ERM proteins [62], ERM proteins are not activated by the rho GTPase-ROCK pathway in astrocytes. 

Apart from the spinal cord lesion studies by Erschbamer et al. [67], however, investigations on astrocytic rho family proteins were carried out only in cell culture [61]. This is well-known to represent a form of glial activation, which involves stimuli and signalling not present in the normal astrocyte. Rho family GTPases in astrocytes in the brain are expressed only at low levels [61,66], and their possible role or link to ezrin/radixin in physiological conditions will have to be tested.

#### 2.3.3. GSK2 

GRK2 is the best-characterised member of the G protein-coupled receptor (GPCR) kinases (GRKs) family of proteins. In their agonist-occupied state, GPCRs allosterically activate GRKs. In turn, the GPCR, as a GRK substrate, is phosphorylated leading to GPCR desensitisation. For GRK2, several additional proteins other than GPCRs, membrane-bound or cytosolic have been described as substrates [42]. Astrocytes in cell culture and in brain express GRK2 [41,71,72]. GRK2 in astrocytes is known to mediate chemokine receptor signalling via ERK1/2 [71], and in a cell line (Hep2 cells) it may also participate in GPCR dependent activation of ezrin [42]. The regulatory phosphorylation site responsible for maintaining ezrin in its active conformation, the C-ter threonine, represents the principle site of GRK2-mediated phosphorylation. It is this phosphorylation that may serve to link GPCR activation to cytoskeletal reorganisation, since inhibition of GRK2 prevents ezrin phosphorylation and membrane ruffling (in Hep2 cells) [42]. Moreover, GRK2-deficiency in GFAP-GRK2(+/−) mice, which have a 60% reduction in astrocytic GRK2, reduces phosphorylation of the GRK2 substrate ezrin [41]. Thus, GRK2 serves not only to desensitise GPCR but also to transduce GPCR-mediated signals in the astrocyte. Together with the data mentioned above, these GRK2 findings in astrocytes further underline the possible link between mGluRs 3/5 activation to glutamate-induced PAP motility [22]. Interestingly, the inhibition of rho kinase or PKC did not abolish GRK2-mediated ezrin phosphorylation in Hep2 cells [42]. This absence of a PKC effect on ezrin phosphorylation [42] may be cell type specific; it is at variance with the results of Burgos et al. [57], who find a strong effect with PKCε in astrocytes. It will be interesting to check whether GRK2 is preferentially localised in the PAPs in the brain, and whether in the GFAP-GRK2(+/−) mice [41] PAPs remain unaltered or are diminished due to reduced ezrin phosphorylation.

#### 2.3.4. Further Candidate Kinases

Another ERM kinase in cell-free assay and in cell culture is lymphocyte-oriented kinase (LOK [73]), but the presence of LOK in neural cells has not been investigated, and in a LOK knock-out mouse no indications of neurological symptoms have been reported [74]. However, LOK appears to play a role in localised, apical phosphorylation critical in forming membrane protrusions like microvilli or glial PAPs [75]; see below).

The human kinase MAP4K4, also termed HGK (for hepatocyte progenitor kinase-like/germinal centre kinase-like kinase), is a member of the human STE20 family of serine/threonine kinases and is the ortholog of mouse NIK (Nck-interacting kinase, not to be confounded with the NF-κB-inducing kinase, NIK) [76]. NIK regulates growth factor-induced lamellipodium formation and phosphorylation of all ERM proteins [77], and this signalling mechanism entails direct phosphorylation of ERM proteins by NIK [77]. MAP4K4 is highly interesting in the context of glioma. Its activity is only regulated at the transcriptional level. Knockdown of MAP4K4 expression inhibited glioma cell migration [78]. Glioblastoma multiforme expresses a characteristic epidermal growth factor receptor (EGFR) mutant (EGFRvIII, de 2–7) that signals constitutively, and is more tumourigenic than the wild-type receptor. Expression of EGFRvIII, but not wild-type EGFR, in a glioma cell line results in the specific up-regulation of a small group of genes, amongst them MAP4K4 [79], all of which influence signalling pathways are known to play a key role in oncogenesis. Wright et al. [76] found MAP4K4 to be upregulated in most tumour cell lines relative to normal tissue, and that expression of an inactive mutant MAP4K4 also inhibited the anchorage-independent growth of cells. Expression of MAP4K4 mutants had a striking effect on growth factor-stimulated epithelial cell invasion so that MAP4K4 appears to play an important role in cell transformation and invasiveness [76]. It remains to be studied whether the MAP4K4 role in glioma malignancy is mediated through ERM phosphorylation, as demonstrated for NIK (see above, [77]). 

### 2.4. Phosphatases

ERM proteins are constitutively both phosphorylated and dephosphorylated in cultured adherent and non-adherent cells, and treatment with calyculin-A, a general protein phosphatase inhibitor, dramatically augments phosphorylated ERM [80]. The major serine/threonine protein phosphatases in eukaryote cells, i.e., those relevant for C-terminally phosphorylated ERM, belong to PP1 and PP2A families [81,82]. In astrocytes, phosphatase(s) acting on C-terminally phosphorylated ezrin or radixin have not been identified, to the best of our knowledge. However, several phosphatases dephosphorylating ERM proteins are known in other cell types; and a few will be mentioned since they may be worthwhile studying in astrocytes as well. 

In a study on cell size control by ezrin, in MDCK II cells, synaptotagmin-like protein 2-a (Slp2-a) interacts with protein phosphatase 1β (PP1β), which is thus recruited to the plasma membrane and inactivates ezrin [83]. Interestingly, the ERM protein dephosphorylation by protein phosphatase 2A (PP2A) modulates mast cell degranulation, an effect mediated by p21 activated kinase 1 (Pak1). This correlates with impaired systemic histamine release in Pak1(−/−) mice and impaired degranulation in ezrin disrupted primary mast cells [84]. 

Ezrin is recognised as having a proactive role in cancer metastasis through control of its phosphorylation status, often by diminished phosphatase activity. Thus, in human laryngeal epithelial cells (Hep-2), altered protein level and reduced activity level of PP2A leads to an increase in ezrin phosphorylation, reorganises the cytoskeleton and promotes cell migration [85]. Similarly, in A549 human lung adenocarcinoma cells, PP2A inhibition results in hyperphosphorylated ezrin and rearrangements of filamentous actin [86]. PP2A activity is also decreased by resistin, an adipocyte-secreted factor known to be elevated in breast cancer patients. Since resistin consequentially increases ezrin phosphorylation, and ezrin knock-down blocks resistin-induced breast cancer cell invasion [87], PP2A-mediated ezrin phosphorylation may be a critical regulator of breast cancer metastasis. 

Phosphatase of regenerating liver (PRL) proteins are a group of protein phosphatases representing candidate cancer biomarkers [88]. An ezrin phosphorylation site not considered so far is tyrosine 146. Inhibition or knock-down of PRL-2 in A549 cells increases the phosphorylation of ezrin on tyrosine 146 and inhibits cell migration and invasion [88]. PRL-3 is over-expressed in the highly invasive HCT116 colon cancer cells, where it dephosphorylates ezrin at the C-terminal Thr567, as a means through which PRL-3 exerts its function in promoting tumour progression [89,90].

ERM phosphorylation state is also equilibrated by the bioactive sphingolipids ceramide and sphingosine 1-phosphate. Plasma membrane ceramide induces ERM dephosphorylation whereas sphingosine 1-phosphate induces phosphorylation. In this mechanism, which is independent of membrane-bound PIP_2_, the plasma membrane ceramide-induced ERM protein dephosphorylation is mediated by protein phosphatase 1α (PP1α) [91]. Ceramide-PP1α-mediated ezrin dephosphorylation is blocked in cancer cells by the widely used anti-tumour agent cisplatin, which also induces an elevation in the activity of the ceramide producing enzyme acid sphingomyelinase, and its redistribution to the plasma membrane [92]. This cisplatin effect induces ezrin relocation from membrane protrusions to the cytosol, loss of lamellipodia/filopodia and appearance of membrane ruffles [92]. 

Future studies may reveal possible roles of ezrin or radixin phosphatases in the physiology of perisynaptic astrocytic ensheathment, glial activation (states), motility of activated astrocytes, and glioma invasiveness.

## 3. Physiological Functions of Astrocytic ERM Proteins 

Although ezrin deficient mice have been generated [93,94], and their aberrant intestinal microvillus formation documented [93], the young pups are hard to raise post-weaning, rendering it difficult to study CNS malformations or functional deficits. In the retina of these new born mice [93], microvilli of Müller cells and retinal pigment epithelial cells are reduced [95]. Applying improved handling and feeding, these mice can survive up to eight weeks [96], but the CNS has not been studied.

### 3.1. Ezrin Localisation and Astrocyte Polarisation 

The relation between the preferential localisation of ezrin and its activation is intriguing. Ezrin has been recognised from the time of its discovery as a preferential marker of surface epithelia, where it is highly enriched in the apical microvilli [14]. With regard to the CNS, ezrin was later localised in the microvilli of retinal Müller cells [10,30], and in the fine processes but not the perivascular endfeet of astrocytes [13], which can be regarded as a basal part of the glial membrane. Thus, ezrin and radixin are preferentially localised at the apical cell pole of astrocytes. In cultured astrocytes, ezrin distributes all over the cell, but its activated form only to the filopodia, and in the brain activated ezrin is preferentially in the PAPs next to synapses [22]. Obviously, this restricted localisation of C-terminally phosphorylated ezrin is linked to the mechanism(s) of ezrin activation. In the early differentiation of intestinal epithelial cells, the apical localisation of PKCι is a key feature in polarisation [19], although other kinases (e.g., PKCα or Akt2) are present at the tip of intestinal villi. It may be concluded that any mechanism phosphorylating ezrin at the C-ter threonine must operate exclusively at the apical domain to achieve full apical localisation of ezrin [19]. In addition, considering that ezrin binding to PIP_2_ is a prerequisite for C-ter threonine phosphorylation (see above), an apical gradient of PIP_2_ versus basolateral PIP_3_ has been suggested as essential for the polarisation of epithelial cells [19,97]. In microvilli, localised phosphorylation is then achieved through an intricate coincidence detection mechanism that requires the membrane lipid PIP_2_ and the apically localised ezrin kinase, lymphocyte-oriented kinase (LOK, also known as STK10) or Ste20-like kinase (SLK) [75]. A recent model of ezrin recruitment to the apical membrane or into filopodia [23] focusses on I-BAR domain proteins, which accumulate at negative membrane curvatures and induce PIP_2_ clusters. It has been proposed that the interaction of ezrin with I-BAR domain proteins is required to enrich ezrin at membrane protrusions, such as filopodia or microvilli [23]. 

For the analogous polarisation or definition of membrane domains in astrocytes [10], it will be important, therefore to establish the subcellular localisation of the candidate ezrin kinases and proteins mentioned above (PKCε, PKCι, LOK, MAP4K4). This will impact the understanding of activity-driven plasticity of glial processes. This may also shed a light on malignancy since the underlying mechanisms of restricted kinase localisation are closely related to astrocyte polarisation, and the breakdown of apico-basal orientation is a hallmark of glioma cells.

### 3.2. Motility of Glial Processes at the Synapse 

The functions of ezrin and radixin in astrocytes may vary depending on the brain region and physiological parameters studied, and the functions of the proteins within the glial cell. The definition of ERM proteins as microvillus proteins is linked to their membrane-to-cytoskeleton link, and phosphorylation at a specific site (see above). However, there may be additional functions based on other interactions and mechanisms. 

The function of ezrin in PAPs in the brain is linked to its membrane-to-cytoskeleton linker function since PAPs consistently display the activated, C-terminally phosphorylated form [22]. Glutamate-induced, activity-associated glial plasticity has been found in several CNS regions (for review see [98]). Stimulation of the glial mGluRs 3 and 5 leads to filopodia dynamics in cultured astrocytes [22]. Applying ezrin immunostaining, PAP dynamics were monitored in the suprachiasmatic nucleus (SCN) of the hypothalamus. Here, the central pacemaker of circadian time is synchronised to the day/night rhythm by the terminals of the retinohypothalamic tract, which release glutamate in a day/night rhythm. Ezrin was found to vary in the same temporal pattern [22], supporting a functional role of ezrin in activity-induced PAP plasticity (see Figure 1). 

### 3.3. Receptor Internalisation 

In a non-neural cell line, β2 adrenergic receptor stimulation has been shown to lead to receptor internalisation, a process mediated by GRK activity and blocked by the inhibition of ezrin function [42]. It has not been investigated to our knowledge whether this applies also to the astrocyte, in particular to the glial mGluR subtypes, mGluR3 and 5. In astrocytes, the regulation of filopodia elongation and retraction, by mGluR3 and/or 5 stimulation is known to be complex and variable [22]. Intracellular signalling of these mGluRs also displays cross-talk [99]. Such GPCR-mediated and ezrin dependent receptor internalisation might contribute to the opposing effects of mGluR stimulation in the astrocyte [22]. In this case, mGluR dependent ezrin activation would mediate both, cytoskeleton motility and receptor internalisation. 

### 3.4. Enhanced Glutamate Transport: Ezrin Link to GLAST/EAAT1 and GFAP 

Ezrin also interacts with the astrocyte cytoskeletal protein GFAP, although it could not be established whether this is direct or mediated by an undefined third protein [100,101], especially since protein interaction was determined from transfected COS cells. GFAP, in turn, binds to the glutamate-aspartate transporter GLAST (also termed EAAT1) and the Na^+^/H^+^ exchanger regulatory factor (NHERF1, previously termed ezrin binding protein 50 (EBP 50) [100], in transfected COS cells [101]). Like ezrin, NHERF1 is restricted to astrocytes throughout the brain [100], it also co-distributes with GLAST; this protein interaction is direct and both proteins localise to all astrocyte processes including the PAPs. Altogether, ezrin is likely to act as a linker to join the NHERF1–GLAST complex to the actin cytoskeleton [101]. The physiological relevance of these interactions is the increased glutamate transport across the cell membrane in the presence of GFAP and NHERF1, in primary astrocytes [101]. The astrocytic cytoskeleton plays an important role also in protecting the brain against glutamate-mediated damage occurring in hypoxic insult. At the cellular level, expression of GFAP is essential in retaining GLAST in the plasma membrane of astrocytes after hypoxic insult (neonatal pig brain, [101]). The actual role of ezrin in this link has remained undefined. The protein complex containing GFAP, GLAST and ezrin also remains paradoxical topographically, considering the authors’ remark [100] that in light microscopy GFAP is expressed in the core of the astrocyte stem process, whereas GLAST localises to the plasma membranes of the astrocytes. However, such a connection has also been demonstrated in another model, the spontaneously immortalised human retinal Müller cells (MIO-M1), which displays neural stem cell characteristics [102]. Stimulation of the NMDA receptor in these cells leads to an increase in the transporter activity of GLAST, mediated in part by NMDA-dependent association of GLAST with the proteins ezrin and GFAP [103]. Here, GLAST co-immunoprecipitated with ezrin, which was phosphorylated at the Tyr354 site [103].

### 3.5. Conclusions

ERM proteins in astrocytes in the brain are key in establishing and maintaining the fine morphology of PAPs. Through regulation of its phosphorylation state ezrin plays a functional role in synaptic activity-induced PAP plasticity although kinase mechanisms remain to be investigated. Ezrin is also involved in the regulation of glial glutamate uptake though not yet fully understood participation in a protein complex comprising GLAST, NHERF1, and GFAP. A possibility indicated in other cell types, and worth studying in astrocytes is the ERM mediated internalisation of integral membrane proteins such as mGluRs or GLAST, known to be regulated also by internalisation and re-integration.

## 4. Ezrin in Neuropathology

### 4.1. Association with Gliomas and Involvement in Molecular Malignancy Mechanisms

The cancers of the CNS comprise a heterogeneous group of tumours most commonly arising from the glial cells of the brain. These gliomas are further divided into astrocytomas, oligodendrogliomas, oligoastrocytomas, ependymomas and other gliomas. The most frequently occurring astrocytomas are graded for their increased malignancy from grade I to grade IV using the WHO classification, with the glioblastoma as the most malignant grade IV astrocytoma [104]. Patients with glioblastomas survive rarely for more than three years. These tumours are characterised by high mitotic activity and extensive migration of tumour cells along the brain capillaries invading the brain diffusely. Hypoxia, necrosis and microvascular proliferation are further characteristics of glioblastoma. However, metastasis of glioblastoma outside the brain is extremely rare. Glioblastomas contain at least two distinct subpopulations of tumour cells: a migratory stem cell/mesenchymal cell-like small tumour cell population, and the typical glioblastoma cells with their often bizarre shape and high mitotic index. Proliferation and migration appear mutually exclusive. It is currently assumed that these subtypes have a common progenitor. Yet it is currently unclear if the tumour cells can alter their differentiation state depending on their environment, and if yes how they can do so [104,105,106].

Due to its multiple functions as a cytoskeleton linker and a signalling protein, the role of ezrin in gliomas has been researched extensively for the last two decades. A potential association of ezrin with malignancy was first demonstrated in transformed fibroblasts [107]. Then, a related ERM protein, schwannomin/merlin was shown to be involved in the development of Neurofibromatosis II related tumours [108]. Our own work has demonstrated for the first time immunohistochemical staining of normal human astrocytes for ezrin [109]. In biopsy specimens from 115 cases, ranging from WHO grade I pilocytic astrocytoma to WHO grade IV glioblastoma multiforme, a significant correlation of ezrin staining intensity with increasing malignancy of astrocytic tumours was found [109]. 

Negative effects of ezrin-expression on the prognosis of malignant tumours were also demonstrated in uveal malignant melanoma of the eye [110]. Since then, ezrin upregulation has been associated with poor prognosis in numerous cancer types [111], including the association of ezrin with malignancy, and poor prognosis in glioma [112,113]. Overall effects of ezrin on the prognosis of carcinomas were found to be statistically more stringent in Asian populations as compared to patients of Caucasian descent [113] indicating a link with other, probably genetically determined factors which will require further research. 

The results concerning astrocytomas were similar in all case groups studied [109,112,113]. However, some discrepancies regarding results for oligodendroglial tumours were evident [109,112,114], confirming the current interpretation of WHO classification [104] that tumours with exclusively oligodendroglial morphology may differ genetically from oligoastrocytomas. Thus, in pure oligodendroglioma, combined IDH-1 mutation and loss of heterozygosity (LOH) 1p19q is associated with better prognosis. A direct association with the low expression of ezrin has not been studied yet. In oligoastrocytomas, ezrin upregulation also correlates negatively with LOH 1p19, where LOH 1p19q is associated with a better prognosis as well [112]. However, it remains unclear whether these traits are directly connected.

High mitotic index, invasiveness and resistance to hypoxia and apoptosis are hallmarks of malignant grade IV glioblastoma. Neuropathological findings suggest a major role for ezrin in the development and progression of malignant glioma which has been associated with several distinct functions of ezrin [115]. Ezrin has been implicated in the organisation of cell shape and migration as well as in the proliferation of glioma cells [105,115,116,117]. However, many isolated observations have been described in glioblastoma cells. Integrating data from other tumour species may help to gain more systematic insight, although their often different functional phenotypes have to be regarded with a caveat. 

Ezrin serves as an interaction partner of the adhesion molecules of the surface such as ICAM-2 and CD44 via the mediation of PIP_2_ [108,115,116] as opposed to PIP_4/5_ which appears to be involved in reactive astrocytes [118]. 

Ezrin expression is upregulated by oncogenic transcription factors such as Myc, and downregulation of tumour suppressor factors such as NF2 (for details see Reference [116]). This results in a disruption of cell–cell contacts through the interaction of ezrin with Fes kinase, and reduced β-catenin levels, which promotes PKC- and CD44-induced cell migration [115,116]. 

ERM proteins are also part of a molecular switch that signals cellular growth or growth inhibition [119]. The association with CD44 is required for the action of merlin, a protein related to the ERM family, on cell proliferation. In logarithmically growing cells, however, merlin has no influence on cell growth, is phosphorylated, and is in a complex with the ERM proteins ezrin and moesin. This complex is also associated with the cytoplasmic tail of CD44. The CD44 ligand, hyaluronic acid, as well as antibodies that mimic CD44 ligands can imitate the effects of high cell density in logarithmically growing cells, rapidly inducing dephosphorylation of merlin, and inhibiting cell growth in a rat schwannoma cell line [119]. PKCε expression, which promotes proliferation was found to be elevated by between three to 30 times in glioblastoma multiforme cells as compared to the levels in normal human glial cell cultures [120,121]. A connection of ERM Proteins to a potential regulation of vascular function involving PKCε has been established in lung tissues [122]. The direct influence of ezrin or PKCε on microvascular proliferation in glioblastoma has not yet been studied.

The actin cytoskeleton and cell adhesion are regulated through cyclic phosphorylation and dephosphorylation of ezrin [18]. In glioblastoma, this system is skewed towards ezrin activation [18,115,117,123,124]. Several protein kinases have been shown to participate in this process. RhoA and Rock signalling pathways have been described to be associated with Ezrin activation in astrocytes and different tumour cell lines, inducing reorganisation of the actin cytoskeleton (see above [63,125,126,127]).

Hepatocyte growth factor (HGF) and BCL-2 family proteins promote invasion via TGF-beta and activity of matrix metalloproteinase (MMP) in human malignant glioma cells using a furin-dependent pathway. The pro-invasive properties of TGF-beta require furin-dependent MMP activity [128]. In particular, HGF may play a specific role in glioblastoma. Recent data have attributed glioblastoma multiforme invasive phenotype to a migratory glioma stem-like cell (GSC) population which are likely to be stimulated by hypoxia [106]. This population consists mostly of small cells with only minimal processes which show strong expression of ezrin [109,129].

Similar to TGF-beta, HGF, an upstream interaction partner of ezrin uses furin-like MMP activity [128] and has been involved in cytoskeleton assembly and reorganisation [130,131]. However, its role in regulating cytoskeleton dynamics is still not fully elucidated. HGF, among other functions, regulates actin-surface-interactions by the interaction of ezrin and FES kinase, which phosphorylates ezrin at tyrosine 477, resulting in the scattering of tumour cells [132]. Thus, ezrin overexpression promotes a migratory phenotype of glioblastoma [117]. Sodium-hydrogen exchanger isoform 1 (NHE1) plays a role in survival and migration/invasion of glial tumour cells in a hypoxic and naturally acidic microenvironment. NHE1 activity is significantly elevated in gliomas. It colocalises with ezrin in lamellipodia and participates in maintaining a more alkaline microenvironment [133]. 

There are additional, isolated results indicating a role of ezrin in glioma malignancy. Thus, NHE1 inhibition prevents ERK activation, and accelerates apoptosis induced by temozolomide, the most frequently used cytostatic medication in glioblastoma therapy [133]. Ezrin also participates in the regulation of phago-lysosomal fusion [134], which might be important for the transport and turnover of therapeutically used substances, or for autophagy in glioma cells [135] Jawhari et al. 2016. The anti-tumour effects of the plant substrate 15α-MP against glioblastoma and breast cancer progression [136] are suggested to be mediated the modulation of Stat3, CyclinB1, Alk, ezrin, merlin, and Erk1/2 functions. A potential interaction between ezrin and the transcription factor Sox2 may be relevant for the sensitivity of glioblastoma cells to ionising radiation [137]. Thus, further research may provide additional targets for therapeutic intervention in the context of ERM proteins.

### 4.2. Ezrin in Epilepsy

Apart from high-grade malignant gliomas, ezrin may play an important role in low grade hamartomatous glial tumours as well. Epilepsy can be caused by low-grade hamartomatous tumours of mixed origin, malformations [104,138] as well as cerebral scars. However, in these cases, there is no clear correlation of ezrin expression with malignancy. Here, Ezrin may offer some insight into the developmental history of these tumours with regard to radial glia and developing neurons, which display ezrin immunoreactivity during normal development (see above [139]). For example, gangliogliomas (GGs) and dysembryoplastic neuroepithelial tumours (DNTs) are mostly benign CNS tumours (WHO grade I) with hamartomatous components and represent the most frequent types of neoplasms in paediatric epilepsy. Available data suggest a pathogenetic relationship between GGs and other glioneuronal malformations of cortical development (MCDs) [138]. Strong ezrin immunoreactivity was observed in the glial component of the mixed glioneuronal GGs and in some of the aberrant neurons [140] including activation of the Pi3K-mTOR signalling pathway. Moreover, ezrin immunoreactivity is not enhanced in DNTs, which consist mainly of dysplastic neuronal cells, suggesting a difference in pathogenetic origins of DNTs and GGs [141].

Ezrin is also overexpressed in a population of abnormal glioneuronal cells in cortical tubers of tuberous sclerosis, a mixed glioneuronal malformation often associated with epilepsy [139].

Other forms of epilepsy cannot be associated as clearly with tumours or malformations, such temporal lobe epilepsy (TLE)/hippocampal sclerosis which may in part be caused by malformation (Focal cortical dysplasia, FCD) and in part by reactive damage due to previous local inflammatory disease (see below). Aberrant expression of ERM proteins was observed in FCD and GG [142]. These alterations may be involved in aberrant phosphatidylinositol 3-kinase (PI3K)-pathway signalling in epilepsy-associated malformations [142].

The differentially expressed gene set in sclerotic hippocampi reveals changes in several molecular signalling pathways, including those associated with astrocyte structure (GFAP, all ERM proteins, paladin), calcium regulation (S100β, chemokine (C-X-C motif) receptor 4) and blood–brain barrier function (aquaporin 4, chemokine (C-C- motif) ligands 2 and 3, plectin 1, intermediate filament binding protein 55 kDa) and inflammatory responses. Immunohistochemical studies show that there is altered distribution of the gene-associated proteins with upregulation of proteins related to activation in astrocytes from sclerotic foci compared with non-sclerotic foci including increased ezrin-IR in astrocytes of sclerotic areas [143,144].

### 4.3. Ezrin Plays a Role Glial Activation and Distinguishes Two Glial Activation States

The experimental paradigms for the study of primary astrocyte shape changes are a conversion from flat to stellate astrocytes (glial stellation), and induction of filopodia formation and motility [145] involving cytokines and other stimuli, and intracellular signalling often not present in normal astrocytes [146]. 

Activation of astrocytes comprises heterogeneous phenomena ranging from plasticity associated changes to scar formation, neuroprotection, and neurotoxic inflammation [146,147,148]. Accordingly, activation of astrocytes occurs in many pathological processes such as scar formation after traumatic lesions, hypoxia, infection, epilepsy or dementia. The morphological alterations of activated astrocytes are heterogeneous as well and range from barely visible alterations to thick fibrous networks. In general, astrocyte activation leads to a more compact cell body and shorter processes with stronger branches [147,148]. This phenotype is rather similar to the morphology of lower-grade astrocytoma cells. It is predominantly seen in activated astrocytes displaying increased ezrin/ERM-IHC [149,150]. Ezrin can be shed into the cerebrospinal fluid (CSF) after experimental traumatic brain injury (TBI) in rats. Ezrin is most likely actively released from periventricular cells since other ERM proteins and β-actin show different expression patterns in these conditions. Ezrin was also found in human CSF after traumatic injury [151]. The mode of release is still unclear and could suggest a potential function of extracellular ezrin.

It may be assumed that ezrin is mechanistically involved in the structural changes accompanying astrocyte activation; however, it has not been systematically studied whether and how ezrin redistributes subcellularly in glial activation, or how its phosphorylation state changes. In neuropathological specimens, ezrin appears to relocate to larger processes in activated astrocytes, especially those surrounding small vessels (see above). The significance of this process is not yet clear, however, one might speculate that it is associated with the described A2 proinflammatory phenotype of activated astrocytes.

Activation of astrocytes, in general, has been demonstrated to be linked not only to a direct response to injury but also to increased plasticity of the contralateral hemisphere after subtle brain injury by experimental differentiation [147]. These processes lead to upregulation of thrombospondin 4, which is attenuated by the absence of GFAP and vimentin [147]. ERM-proteins can induce an integrin-dependent migration of activated astrocytes after trauma [152]. However, in this study, the specific ERM protein family member involved was not identified. Ezrin and GFAP can be expressed differentially after TBI [153]. Upregulation of ezrin in combination with downregulation of GFAP in perilesional activated astrocytes after exposure to environmental enrichment prior to a cerebral stab wound was interpreted as a neuroprotective effect of ezrin related to the promotion of synapse formation [153]. 

Neuroprotective effects of ezrin were seen in other studies as well: Ezrin may play an integral part in the recovery from hypoxic brain insult [154]. Urokinase-type plasminogen activator (uPA), which was shown to promote neurite growth during development also induces the expression of ezrin in astrocytes in a hypoxic environment. This resulted in the formation of PAPs that entered in direct contact with the injured synapses protecting them from degradation [155]. UPA binding to activated astrocytes also led to the release of thrombospondin-1 which in contact with synaptic low-density lipoprotein receptor related protein 1 (LRP1) promoted synaptic recovery [156].

The functions of ezrin may also be ambivalent in different environments: Ezrin also mediates downstream effects of Stat3 signalling in astrocytes which leads to increased scar formation of reactive astrocytes [157]. Inhibition of RhoA by STAT3 involves ezrin which is known to downregulate RhoA activity in other cells [157]. However, how the loss of Stat3 results in reduced phosphorylation of ezrin at threonine residues is currently unclear. The role of ezrin in neuroinflammation may be further elucidated by studies investigating IL-1β in reactive human astrocytes: IL-1β induces a reactive astrocyte phenotype by deactivation of the Rho-GTPase-Rock-axis. Thus, IL-1β leads to filopodia formation in cultured astrocytes by simultaneous action on f-actin, ERM-phosphorylation, and ICAM-1 induction [158]. ICAM-1 serves as an interaction partner of ezrin with which it is colocalised in astrocyte filopodia [159]. Through interaction with different partners such as PIP_4/5_ as opposed to PIP_2_, ezrin may rather promote neuroinflammatory processes [118].

Ezrin also regulates endothelial cell proliferation and angiogenesis via TNF-α-induced transcriptional nuclear repression of cyclin A [160]. This might be relevant for neuroinflammation as well as for microvascular proliferation in glioblastoma, which has not been studied yet. TNF-α leads to Rho kinase-mediated phosphorylation of ezrin and to CXCR4-activated glutamate release by astrocytes, which amplifies microglia-triggered neurotoxicity [161] and may also play a role in Alzheimer dementia-associated pathology [162]. A further connection of (not specified) ERM proteins to CXCR4 has been demonstrated for neuroblast migration after TBI [163]. 

We have shown a correlation of immunohistochemical up-regulation of ezrin with the severity of HIV encephalopathy but not with cerebral HIV-infection per se [149]. A direct interaction of ezrin with HIV-gp120 has been described [164,165], possibly related to HIV-associated altered glutamate uptake of astrocytes, and to neuronal toxicity through induction of apoptosis [166] or CXCR4-activation [165]. However, it appears more likely, that ezrin up-regulation in HIV-encephalopathy is associated with a specific state of astrocyte activation [149]. This specific reactive phenotype of astrocytes is linked to severe CNS disease, loss of physiological function and a probably irreversible alteration of astrocyte morphology [159]. Subsequently, such a potential link of ezrin to harmful astrocyte activation has been corroborated by further work ([118,161]; see above). Similarly, induction of a senescent state by CDK5-mediated phosphorylation of ezrin was shown in a SAOS-2 sarcoma line [167]. 

The concept of two different types of astrocyte activation has been expanded since then. Two different groups of reactive astrocyte populations have been identified using transcriptomic data from tissues of elderly people [168,169]. A1 astrocytes are regarded as cytotoxic and show increased expression of genes related to interferons and other proinflammatory mediators belonging for example to complement signalling and acute phase response signalling, the complement system and the antigen response including upregulation of Stat3 [169]. A2 astrocytes are regarded as neuroprotective and display upregulation of genes regulating signal transduction, plasticity and neuroprotection, such as SSH and the ERM proteins and CD44. Both astrocyte types are present in the brains of elderly people in varying ratios [168]. Especially the upregulation of ezrin has been interpreted as being beneficial in this context as a promotion of neuroprotection [169]. However, it has also been noted that RhoA is not elevated but, instead, rather more unusual interaction partners of ERM-proteins, such as RhoU and RhoJ [169] which may point in a different direction. Through interaction with Stat3 and TNF-α among other interaction partners upregulated in A1 astrocytes, ezrin may also have pro-inflammatory functions [170,171] as well. Alternatively, the clinical evidence in cerebral HIV-infection may also point towards exhausted neuroprotective attempts in the face of growing dysfunction. Further research will be necessary to determine more precisely, how ezrin may participate in both neuroprotective and neurotoxic activities and whether or how such a switch may be effected. 

### 4.4. Conclusions

Altogether, astrocytes are key players in many disease processes of the CNS. It has been found that ezrin is associated with most of them in a disease-specific manner [25]. On practical terms, relatively low-cost immunocytochemistry for ezrin has been proven highly useful for the evaluation of CNS disease. This ranges from diagnosis of glial tumour type and malignancy to prognostic assessment of these entities; although for this application, combination with molecular pathological methods may be necessary to establish LOH 1p 19q and IDH-status [172] in addition to other molecular tumour markers. It is likely that the alteration of ezrin expression and redistribution within the tumour cells represents a pivotal element in the development of highly malignant and aggressive glioma. However, further research will be necessary to better establish its role and that of its interaction partners. Due to the ubiquitous presence of ezrin throughout the body, the search for therapeutic targets will have to focus on interaction partners with higher organ specificity. First results also point towards an important role of ezrin in the development of pro-inflammatory astrocyte activation, allowing for the assessment of the severity of disease in CNS neurodegenerative processes. However, determination of disease specificity and neuropathological assessment protocols will have to be established. Further work is necessary to establish direct links to astrocytic inflammatory processes.

## Figures and Tables

**Figure 1 ijms-20-03776-f001:**
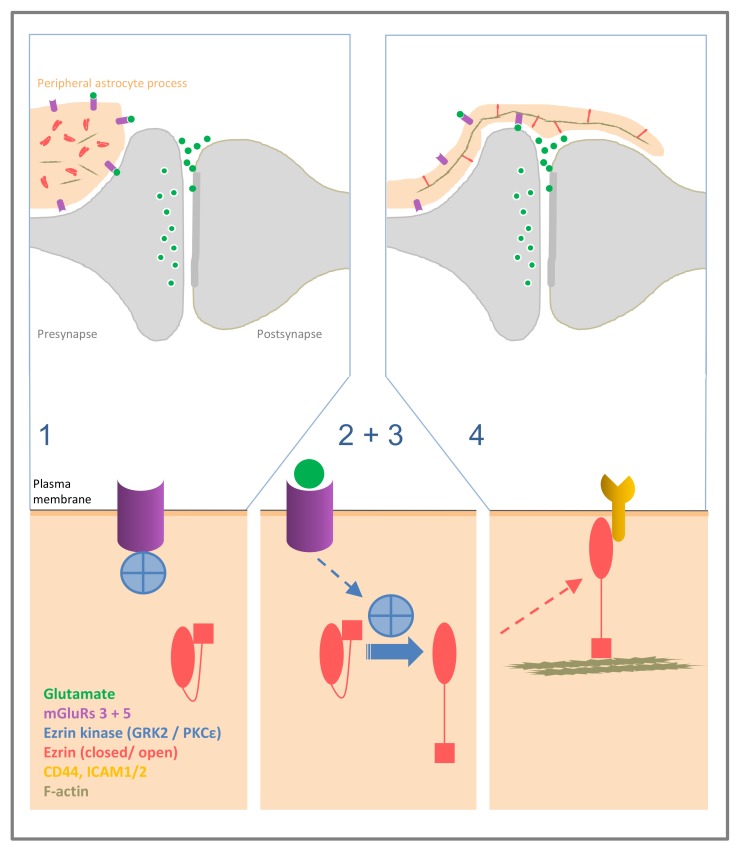
Possible mechanism of elongation of peripheral astrocyte processes at the synapse. Larger glial processes in the vicinity of a synapse contain ezrin and radixin in its closed, inactive form, and display mGluR5 and inactive ezrin kinases, such as GRK2, associated to the receptor, or PKCε (step **1**). Binding of synaptically released glutamate to glial mGluR 5 leads to dissociation of GRK2 from the receptor, or activation of PKCε (**2**), which in a subsequent step (**3**) leads to C-terminal phosphorylation of ezrin/radixin. Activated ezrin/radixin leads to morphological changes (**4**), by linking membrane-bound proteins to the actin filaments and tethering the glial cell membrane to the actin cytoskeleton; a basis for formation and motility of the extremely fine glial processes.

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
