# Peer review of "Perspectives for Ezrin and Radixin in Astrocytes: Kinases, Functions and Pathology"

_ijms, 2019, doi:10.3390/ijms20153776_

Round 1

Reviewer 1 Report

Abstract

-        ERM proteins: please, specify what ERM refers to

Main Text

-        Line 151: …”although not in the CNS”…

For clarity for readers, the authors could specify that they refer to the Peripheral Nervous System.

-        Lines 157-161: to talk about adult neurogenesis and more specifically about subventricular zone neurogenesis:

I suggest defining explicitly that adult neurogenesis occurs in the subventricular zone and the subgranular zone of the dentate gyrus of the hippocampus.

Additionally, if the authors talk about the rostral migratory stream, they should briefly explain the features of adult neurogenesis in the subventricular zone.  Non-specialized readers may not know what the rostral migratory stream is.

Finally, is there not data on ezrin/radixin in hippocampal adult neurogenesis?

-        Line 165: …”in cultured astrocytes and astrocytes in the brain”… reads a little weird. The authors could just indicate …”in astrocytes”…

-        Line 200: please, speficy what HUVEC means

-        Lines 227-228: PKC epsilon has been assayed to phosphorylate Ezrin in primary astrocytes, but what was the outcome, the result?

-        Line 237: it is a little weird to talk about the assay of PKC epsilon induced phosphorylation of ezrin in a paragraph (lines 227-228), and to talk about the result of the experiment in the next paragraph. I suggest the authors, to link somehow both paragraphs. One possibility could be to add something similar to …”in the next section, we will summarize the PKC isoforms that phosphorylate ezrin in cell lines and astrocytes”…

-        Line 282: the authors state: …”ERM proteins are not activated by the rho-GTPase –ROCk pathway in astrocytes”… Not even in the case of reference 68?

-        Lines 462-464: this was already mentioned in a previous section of the manuscript. Overall, the manuscript is rather long, with many details. I suggest keeping the most important information and shortening all the rest.

I think that the authors have made a big effort to compile and organize all the information exposed in the manuscript. However, in my opinion, there is too much information in the manuscript, it is too long, it has too many details. I encourage the authors to summarize as much as they can, stressing out the important information, so that the understanding of the general idea of the manuscript is more intuitive for the reader. In the current form, the reader gets lost in the details and the abundance of information.

Reviewer 2 Report

The review by Derouiche & Geiger „Perspectives for Ezrin in Astrocytes: Kinases, functions and pathology”, has been extensively revised and important new pieces of information have been added. The authors have addressed all my issues and I do not have further questions. The new added material makes the manuscript clearer and stronger. I think that in its present version the manuscript  is suitable for publication and will be of great interest for glia researchers, as well as for a broader community of neuroscientists studying ezrin proteins in astrocytes during physiological and pathological conditions.

Author Response

The review by Derouiche & Geiger „Perspectives for Ezrin in Astrocytes: Kinases, functions
and pathology”, has been extensively revised and important new pieces of information have
been added. The authors have addressed all my issues and I do not have further questions. The new added material makes the manuscript clearer and stronger. I think that in its present
version the manuscript is suitable for publication and will be of great interest for glia
researchers, as well as for a broader community of neuroscientists studying ezrin proteins in
astrocytes during physiological and pathological conditions.
We thank the Reviewer for improving the MS and the kind appreciation of our work.

This manuscript is a resubmission of an earlier submission. The following is a list of the peer review reports and author responses from that submission.

Round 1

Reviewer 1 Report

In the review „Perspectives for Ezrin in Astrocytes: Kinases, functions and pathology”, Derouiche & Geiger summarize the current knowledge regarding the ERM proteins, putting a special focus on the role of these proteins in the astrocytes, and in the nervous system in general. This manuscript is comprehensive and well structured. However, I find that some issues are a bit unclear or missing. Hence, I have few suggestions which the authors may want to consider.  

(1)  In the nervous system, two major types of astrocytes are distinguished: protoplasmic astrocytes in the grey matter and fibrous astrocytes in the white matter.  I think that in a general review like the present one, this issue deserves attention. It would be very interesting to give information regarding ERM protein expression, localization, and function in protoplasmic vs. fibrous astrocytes, especially because these two types of astrocytes are structurally very different. For instance: the fibrous astrocytes seem to have much less tiny processes than protoplasmic ones. Ezrin is mainly found in the tiny peripheral processes of astrocytes. Does it mean that fibrous astrocytes express less ezrin? Would this have any functional implications for normal or diseased white matter?  

(2)  At the beginning of the manuscript, the authors state that ezrin mainly localizes to the peripheral processes of the astrocytes. This may also be true for GLAST because it takes up glutamate at synapses. In the part “Enhancing glutamate transport: Ezrin link to GLAST and GFAP” (line 427), the authors talk about interaction between ezrin, GLAST, and GFAP. I am wondering in which system this occurs because in the immunohistochemical stainings, the anti-GFAP antibody labels mainly the large processes of the astrocytes. Is GFAP really present in the peripheral processes of astrocytes where ezrin and GLAST are placed, in culture or in vivo? I am also wondering which age of the animals is described because in the adult brain GLAST is expressed in the cerebellar rather than cortical astrocytes (while the latter mainly express the GLT-1). The authors should clarify these issues.

(3)  In the parts: “Ezrin in glial activation” and “Ezrin distinguishes two glial activation states”, the authors talk about changes in the expression of ezrin and changes in the function of corresponding intracellular cascades. In this context, it would be important to indicate more clearly what the sub-cellular localization of ezrin in reactive astrocytes is. Reactive astrocytes largely lack tiny peripheral processes. Does it mean that ezrin specifically relocates/traffics to the thicker primary processes of astrocytes after injury? If this is the case, does this have any functional meaning? Or do the astrocytic processes simply contract upon injury and ezrin moves with them “automatically”? 

(4)  In line 588-589, authors state: “In general, astrocyte activation leads to a more compact cell body and shorter processes with stronger branches. This phenotype is rather similar to oligodendrocyte morphology…..”. I am not sure that similarly between the morphology of reactive astrocytes and oligodendrocytes is really correct and can be stated like this. To the best of my knowledge, oligodendrocytes in situ and in vivo usually have several rather thin processes which terminate with myelin sheathes, and structurally they have little, if any, similarities with reactive astrocytes. The authors should check the literature to verify this issue.

(5)  I think that it is a very nice idea to include a figure into the manuscript. But I find the figure in its present version suboptimal, especially the upper panel. Half of it is occupied by an empty grey postsynaptic terminal and carries little, if any, information. At the same time, it is unclear why mGluRs are located so far from the release sites, where ezrin is located, how glutamate reaches the mGluRs if an astrocytic process is an obstacle, etc. The authors should re-design the figure. It might be a good idea to show the dynamics of the whole process, for instance making 2-3 small sub-panels at the top, showing (a) what happens before glutamate binding, (b) how mGluRs and ezrin are activated upon glutamate binding and how the extension of the astrocytic process is triggered, (c) which cascades, events are triggered subsequently, etc

(6)  Minor point: the abbreviation “ERM” should be explained at the beginning of the manuscript.

Reviewer 2 Report

General comments:

The review describes a lot of information on ERMs (in different cell lines and astrocytes). However, the boundaries of the information obtained from other cell lines and astrocytes are not clear in the way the review is presented. In my opinion, not all the information provided by the review is necessary to understand the possible role of ERMs in astrocytes. I suggest shortening a little bit the review (without losing the important information), but focusing more, and clearly discussing the implications in astrocytes and CNS biology.

For example, for content organization, I suggest the following:

1.       Introduction on CNS, astrocyte biology and description of ERMs in CNS context, mainly focusing on astrocytes (using different paragraphs for each sub-section)

2.       Potential mechanisms of ERM regulation in astrocytes (upstream signals, binding partners, and kinases). I would shorten the description of the different kinases, adding different paragraphs for PKC, small GTPases, and GSK2 (and I would focus mainly on the implications on astrocytes after a brief description in other systems).

3.       Ezrin localization and astrocyte polarization (I don’t know if this section could fit well on section 1 or may be inside the section of physiological function of ERMs in astrocytes). Finally, this information, at least in part, is somehow repeated.

4.       Physiology: a conclusion/discussion on the role of ERMs in astrocyte physiology is missing.

5.       Neuropathology: a conclusion/discussion on the role of ERMs in neuropathology, and astrocyte contribution to that neuropathology is missing. I suggest shortening a little bit this part to 3 subsections: gliomas/malignancy together, epilepsy, glial activation (including the 2 activation states).

Specific comments:

Title:

Perspectives for Ezrin in Astrocytes: Kinases, Functions, and Pathology

-          Is it possible that not only kinase type enzymes regulate the activation of ERMs in astrocytes? For example, phosphatases? Has anyone tried to analyze this possibility?

If this idea makes sense, I suggest modifying the title to “…Activation mechanisms, functions, and pathology”.

Keywords:

-          I think that “ezrin” is missing.

General text:

-          Line 54: “astrocytes are the major representative of the non-neuronal, glial cells in the CNS”. What do you mean with this sentence? Are astrocytes more abundant than other glial cell types? Are they more important than other glial cell types? Please, specify.

-          Line 60-63:  all the general functions described for astrocytes in these lines rely on peripheral astrocyte processes? Or do PAPs somehow participate in these functions? Finally, the astrocyte is not only the PAP, and I assume that other parts of the cell are also important for general function.

-          Line 85: reference for moesin in endothelial cells is missing.

-          Line 87: reference for the localization of ezrin and radixin in astrocytes is missing.

-          Line 95: for clarity, I suggest including the threonin of ezrin in brackets too.

-          Line 98: phosphorylation sites also imply dephosphorylation. Any clues on dephosphorylation for activation of ERMs? It may be interesting to speculate on other possible mechanisms of ERM regulation.

-          Line 119: “or what are the kinases for T567 phosphorylation (in ezrin)”. I way to simplify this issue across the review is to refer to this threonine as the C-ter threonine phosphorylation site (this way, it spans all ERMs).

-          Line 126: I understand that the review focuses mainly on ezrin, however other ERMs are also described sometimes. I suggest that on titles, introductions etc, ERMs are referred. Then, in specific situations, the authors can describe the actual contribution of each of the ERMs, or focus more on Ezrin.

-          Lines 127-133: a lot of sentences are included to indicate that although it was thought that ERMs were not present in the CNS, currently it is assumed that they are. I suggest shortening the text across the review without losing the general idea.

-          Across the review, the authors use the terms “in situ”, “in vitro”, and “in vivo” in a way that it is not clear to me. Please, specify what do you mean with each of this terms. In situ, does it mean a local effect? In vivo, were the experiments performed in animals?

-          Line 198: “CD43 is a leukocyte antigen not detected in astrocytes”. What is the relevance of this sentence at the end of the paragraph?

-          Line 211: at the crucial C-ter threonin phosphorylation site?

-          Line 222: in (primary) astrocytes?... in primary astrocytes or in astrocytes.

-          Line 280: the beginning of the paragraph is a mess, please re-organize and focus on the main message.

-          Line 360: I suggest writing the title as general as possible, just ERM proteins.

-          Line 485: is it in glioma malignancy? I would shorten the text and include this section on the previous one.

-          Line 539: …”phago-lysosomal function”. Is then ezrin expressed in brain professional phagocytes, the microglial cells?